# Role of Gut Microbiota, Probiotics and Prebiotics in the Cardiovascular Diseases

**DOI:** 10.3390/molecules26041172

**Published:** 2021-02-22

**Authors:** Anna Oniszczuk, Tomasz Oniszczuk, Marek Gancarz, Jolanta Szymańska

**Affiliations:** 1Department of Inorganic Chemistry, Medical University of Lublin, Chodźki 4a, 20-093 Lublin, Poland; 2Department of Thermal Technology and Food Process Engineering, University of Life Sciences in Lublin, Głęboka 31, 20-612 Lublin, Poland; 3Institute of Agrophysics, Polish Academy of Sciences, Doświadczalna 4, 20-290 Lublin, Poland; m.gancarz@ipan.lublin.pl; 4Department of Integrated Paediatric Dentistry, Chair of Integrated Dentistry, Medical University of Lublin, Chodźki 6, 20-093 Lublin, Poland; jolanta.szymanska@umlub.pl

**Keywords:** cardiovascular disease, probiotic, prebiotic, gut microbiota, human health

## Abstract

In recent years, there has been a growing interest in identifying and applying new, naturally occurring molecules that promote health. Probiotics are defined as “live microorganisms which, when administered in adequate amounts, confer health benefits on the host”. Quite a few fermented products serve as the source of probiotic strains, with many factors influencing the effectiveness of probiotics, including interactions of probiotic bacteria with the host’s microbiome. Prebiotics contain no microorganisms, only substances which stimulate their growth. Prebiotics can be obtained from various sources, including breast milk, soybeans, and raw oats, however, the most popular prebiotics are the oligosaccharides contained in plants. Recent research increasingly claims that probiotics and prebiotics alleviate many disorders related to the immune system, cancer metastasis, type 2 diabetes, and obesity. However, little is known about the role of these supplements as important dietary components in preventing or treating cardiovascular disease. Still, some reports and clinical studies were conducted, offering new ways of treatment. Therefore, the aim of this review is to discuss the roles of gut microbiota, probiotics, and prebiotics interventions in the prevention and treatment of cardiovascular disease.

## 1. Introduction

In recent decades, the incidence of cardiovascular disease (CVD) increased to such extant that they became the principal cause of death worldwide. This is especially noticeable in high- and intermediate-income countries. However, the complex etiologies of CVD and the incomplete understanding of the underlying mechanisms hampered the development of prevention strategies [1].

An unhealthy diet has long been recognized as a major factor for cardiovascular disease morbidity. A connection between diet and cardiovascular events was established through the determinants of metabolic stress and overweight, i.e., adiposity and the presence of visceral fat [2]. While genetic variation has an important influence on Body Mass Index and the distribution of body fat, environmental factors are believed to make a major but still-to-be elucidated contribution to the variation in obesity between different individuals [2,3]. It was not until recently that the complex interactions between dietary components and intestinal microbiota and their food-produced metabolites were acknowledged to play a role in cardiovascular health. Increasing awareness of the role of the gut microbiome in CVD attracted the attention of researchers interested regarding the potential role of probiotics as a study target, e.g., in the prevention of atherosclerosis and other forms of CVD [3,4]. Although knowledge of the changes in the composition of the microbiome associated with coronary heart disease or atherosclerosis is still limited, there is a growing body of evidence supporting this relationship [5].

Functional foods can have beneficial effects against various risk factors associated with cardiovascular disease. However, little is known about the roles of probiotic and prebiotic supplements as important dietary components in the prevention and treatment of CVD [6]. Therefore, the aim of this review is to discuss the roles of gut microbiota and probiotics supplementation in the development of cardiovascular disease based on available reports and clinical studies.

## 2. Cardiovascular Disease: Risk Factors and the Mechanism of Development

Cardiovascular diseases involve the heart or blood vessels, the most well-known being coronary artery disease, stroke, hypertensive heart disease, cardiomyopathy, venous thrombosis, arrhythmia, and thromboembolic disease. CVD is a growing global health problem [7,8,9]. In 2015, 18 million deaths were attributed to cardiovascular diseases, accounting for approximately one-third of all-cause deaths and representing an increase of 12.5% from 2005. The American Heart Association reports that 92.1 million adults in the US currently have CVD, and it is predicted that approximately 43.9% of the entire U.S. population will have CVD by 2030 [10].

Atherosclerosis is a lipid-driven, chronic, inflammatory disease that is characterized by the formation and progressive growth of atherosclerotic plaques in the walls of arteries. Atherosclerosis is a major predisposing factor for stroke and heart attack. Various immune-mediated mechanisms are implicated in the disease initiation and progression [11]. The initiating incidents of atherogenesis include the retention of lipoproteins in the subendothelial space of the arteries and the activation of endothelial cells. Monocytes enter the vascular wall and differentiate into tissue macrophages. The macrophages ingest lipoproteins and turn into foam cells. Synthetic vascular smooth muscle cells accumulate in atheromas and secrete extracellular matrix proteins. Smooth muscle cells and collagen are important components of the fibrous cap that covers the atherosclerotic plaque [12].

Factors that influence the risk of developing cardiovascular diseases include genetics and a poor lifestyle (lack of activity, unhealthy diet, smoking, alcohol). Hypertension is the most common modifiable risk factor in CVD [13,14]. High blood pressure is often associated with metabolic deregulation, which leads to high blood cholesterol levels that, such as glucose in type 2 diabetes, damage blood vessels and lead to atherosclerosis. The mutual interaction between hypertension and hypercholesterolemia and their influence on the development of atherosclerosis include the renin–angiotensin–aldosterone system and endothelial dysfunction [12,15].

## 3. Gut Microbiota and Its Effects on Human Health

Microbes colonize all surfaces of the human body, but the human gut is the site of a particularly rich microbiome. It is characterized by an ecological diversity of microorganisms, with more than 100 trillion microbial cells living symbiotically within it [16]. Microbial colonization in the gastrointestinal tract starts immediately after birth. In healthy people, the microbiota lives in a symbiotic relationship with the host, influencing host health by regulating the metabolism of nutrients, protecting against pathogens [17], and providing signals to immune cells to improve host physiology and immunity [18,19,20,21].

Research links the gut microbiota to the development of several cardio-metabolic diseases, including obesity, type 2 diabetes mellitus (T2DM), and cardiovascular disease [22,23,24,25]. The microbiome, particularly in the colon, forms a “bioreactor” that ferments food components that have escaped digestion in the upper parts of the intestine (proteins, carbohydrates, and dietary fibers), breaking these into metabolites or microbial products, e.g., short chain fatty acids and secondary bile acids. The gut microbiota can also transform other dietary components, e.g., polyphenols, into potentially metabolically more relevant forms [26]. Hence, it is not surprising that alternations in the gut microbiota composition might play a role in maintaining human metabolic balance and cardiovascular health.

## 4. Gut Microbiota, Its Diet-Derived Products, and Cardiovascular Diseases

Many factors are associated with the increase in risk of CVD [7,8,22], but growing evidence indicates that the intestinal microbiome and metabolites contribute importantly in the progression of such diseases. A study conducted by Cui et al. reported that significant differences in the bacterial composition were found between patients with chronic heart failure and control subjects. Of particular note, a decrease in the level of *F. prausnitzii* and an increase in the level of *Ruminococcus gnavus* were observed in patients with chronic heart failure [27]. Another study showed that patients with heart failure had higher levels of *Prevotella*, *Hungatella*, and *Succinclasticum* and lower levels of the Lachnospiraceae family (*Faecalibacterium* and *Bifidobacterium*) than did control subjects [28]. Moreover, juxtamucosal bacterial overgrowth and higher bacterial adhesion were observed in patients with heart failure [29]. In addition, pathogens such as *Candida*, *Shigella*, *Yersinia*, and *Campylobacter* were found to be increasingly present in the stools of patients with CVD [30]. Finally, Jie and coworkers [31] analyzed the gut bacterial composition in patients with atherosclerotic cardiovascular disease and observed significant differences with higher levels of *Enterobacteriaceae* and *Streptococcus* spp.

Gut bacteria are able to produce diet-derived metabolic products capable of influencing the host’s cardiovascular condition, for example, circulating levels of branched-chain amino acid metabolites, tryptophan, and histidine were associated with insulin resistance and vascular disease. An important example is imidazole propionate, which is formed after the metabolism of histidine. There was a clearly higher concentration of imidazole-propionate in the portal blood of obese diabetes patients compared to obese patients without diabetes [6].

The intestinal microbiome communicates with distant organs, including the heart, through a variety of ways. Among these are the production of trimethylamine (TMA)/trimethylamine N-oxide (TMAO), short-chain fatty acids, bile acids, lipopolysaccharides (LPS), and peptidoglycans [22]. TMAO is one of the more extensively studied metabolites formed by the gut microbiota and comes with a potential role in atherosclerosis. TMA is formed by gut microbiota after meals containing choline, phosphatidylcholine, or carnitine, which are present in foods with high levels of saturated or unsaturated fat. Humans do not possess TMA lyases, so all TMA is formed by the gut microbiota. After absorption, TMA is transported to the liver, where the hepatic enzyme flavin-monooxygenase-3 (FMO3) oxidizes TMA to TMAO [6].

Elevated serum levels of TMAO are positively correlated with early atherosclerosis in humans and monitoring helps to predict mortality risk in patients with stable coronary artery disease and acute coronary syndrome [32,33]. Studies showed that elevated plasma levels of TMAO were associated with the severity of peripheral artery disease and with high risk of cardiovascular mortality in patients with peripheral artery disease [34]. In meta and dose-response analysis studies, elevated plasma TMAO concentrations correlated with increased incidence of major adverse cardiovascular events in coronary heart disease patients [35]. Moreover, increased levels of TMAO were significantly correlated with proinflammatory monocytes in patients with stroke. Haghikia et al. [36] reported that the elevated plasma level of TMAO was also associated with increased cardiovascular events such as myocardial infarction, recurrent stroke, and cardiovascular death. Overall, the mechanisms underlying the effect of TMAO on CVD are not completely explored.

Many studies showed that short-chain fatty acids (SCFA) have a contributing role in CVD. High-fiber diet and acetate supplementation were shown to be able to alter the gut microbiota composition, resulting in the prevention of hypertension and heart failure in hypertensive mice [37]. Nondigestible fibers are fermented in the colon by gut microbiota, which leads to the production of SFCAs, mainly butyrate, propionate, and acetate. The concentrations of SFCAs are lower in patients with atherosclerotic vascular disease or hypertension [6]. SCFAs probably have a beneficial effect on atherosclerotic plaque formation by improving intestinal barrier function [38]. In mouse models of hypertensive cardiovascular disease, propionate attenuated hypertension and its cardiovascular sequelae and reduced the atherosclerotic plaque area and the frequencies of splenic effector memory T cells and splenic T helper 17 cells [6]. SCFAs modulate immune and inflammatory responses via many receptors, e.g., via activation of free fatty acid (FFA) receptors and G protein-coupled receptor 109A and inhibition of histone deacetylases (HDACs) [39]. Since FFA receptors are present in endothelial cells, binding of SFCAs to receptors may evoke not only the stimulation and dampening of the production of inflammatory cytokines, but also influence migration and recruitment of immune cells to the atherosclerotic plaque. SCFA metabolic effects are also mediated via FFA receptors and have direct effects on endothelial cells via HDACs.

It is also recognized that bile acids have a contributing role in CVD. These substances are primarily formed in the liver from cholesterol via two pathways, namely, the classical pathway and the alternative pathway. In the classical pathway, cholesterol is converted into the primary bile acids cholic acid and chenoxycholic acid by enzymes [6]. Bile acids are conjugated with glycine and taurine to form glycocholic acid, taurocholic acid, glycochenoxycholic acid, and taurochnoxycholic acid; these compounds lower pH and the solubility of many nutrients is improved. These bile acids are released into the duodenum, especially after the intake of food, to facilitate digestion and improve the uptake of lipids and lipophilic vitamins [40]. The majority of these substances are reabsorbed in the distal ileum via the sodium-dependent bile acid transporter and returned to the liver via the portal system [41]. The colonic microbiota can convert primary bile acids which are not reabsorbed to secondary bile acids, e.g., deoxycholic acid, lithocholic acid, and other secondary bile acids. The composition of the colonic microbiota has a strong influence on the amount of secondary bile acids formed [6]. In addition to their function in promoting the absorption of lipids and vitamins, bile acids are involved in metabolic processes, intestinal motility, inflammatory processes and liver regeneration. Bile acids exert these effects via bile acid receptors, of which farnesoid X receptor (FXR) and TGR5 receptor are most important and present in many cell types and tissues. Binding of bile acids to FXR reduces lipid levels, improves insulin sensitivity and suppresses hepatic gluconeogenesis, while stimulation of TGR5 reduces the production of cytokines.

In conclusion, the intestinal microbiome plays a vital role in CVD via several routes, and several studies were published relating these diseases to an altered intestinal microbiota structure and function (Table 1). Evidence exists that lifestyle and diet, physical activity, and smoking are modifiers of gut microbiota and, subsequently, are modifiers of cardiovascular health. Hence, therapeutics targeting the specific gut bacteria and probiotics supplementation have promising effects that could be used to treat CVD [22,42].

## 5. Probiotics and Prebiotics: General Information

Probiotics (*Greek*; Pro: promotion, Biotic: life) are defined as “live microorganisms which, when administered in adequate amounts, confer health benefits on the host” [47]. A large range of fermented products, such as yogurt, kefir, sauerkraut, tempeh, and kimchi, which serve as sources of probiotic strains, are part of the human diet across diverse cultures. According to the current state of knowledge, probiotics encompass both bacteria (*Lactobacillus*,* Lactococcus*, *Leuconostoc*, *Pediococcus*, *Propionibacterium*, *Bifidobacterium*, *Bacillus*, some *Streptococcus*, *Enterococcus*, *Escherichia coli*) and yeast (*Saccharomyce*s) genera [48].

Many factors influence the effectiveness of probiotics, including interactions of probiotic bacteria with the host and its microbiome. In order to make a positive impact, probiotics must chemically or physically inhibit the growth of pathogenic bacteria (e.g., *Enterococcus faecalis*, *Salmonella enterica* subsp. *enterica* serotype *Enteritidis*, *Listeria monocytogenes*, *Staphylococcus aureus*, and *E. coli*) by immune, hormonal, and neuronal manipulations. It is important that they also stimulate the growth of beneficial microorganisms [49].

More and more research increasingly claims that probiotics alleviate many disorders related to the immune system, cardiovascular health, cancer metastasis, depression, anxiety, type 2 diabetes, and obesity [48]. Recently, the safety profiles of different probiotics as a function of different genera, species, and strains, coupled with their relevance to diverse individuals or at-risk populations, attracted attention [48]. The FAO/WHO (Food and Agriculture Organization/ World Health Organization) [50] guidelines on probiotic evaluation from 2002 reported that probiotics may theoretically be linked to specific types of side effects in patients with underlying medical conditions. The at-risk population groups are broadly characterized by weakened immune systems, gut dysbiosis, and/or impaired intestinal barriers, therefore, it is important to carefully assess the safety associated with deliberate administration of probiotics.

According to the provisions of the WHO, the number of living cells in probiotic foods at the time of human consumption may not be lower than 106 cells per 1 mL or 1 g of product. Furthermore, the therapeutic dose is 108–109 cells in 1 mL or 1 g of product [51]. Of note, the contained microorganisms must be resistant to the action of gastric juice and bile salts. After passing through this chemical barrier, probiotics should then adhere to the surface of the intestine, where their health-promoting functions can be realized [51].

Products deemed probiotics enhance the nonspecific cellular immune response through the activation of natural killer cells and macrophages and the release of various cytokines. They can also improve the gut mucosal immune system by increasing the number of IgA(+) cells [52]. Moreover, probiotics can aid the process of digestion and the breakdown of lactose, improve the absorption of minerals, and enhance the synthesis of many vitamins (thiamin, riboflavin, niacin, pantothenic acid, vitamin K). They play an important role in the treatment of various diseases, such as hepatic disease, diarrhea, and gastroenteritis. In addition, they were also shown to have antiproliferative, proapoptotic, and antioxidative properties [52].

Prebiotics represent substances most used to maintain a normal gut microbiota and restore its equilibrium when homeostasis is affected [53,54,55]. Prebiotics contain only substances which stimulate microorganism growth; there are no bacteria in their composition [9]. These substances can be obtained from various sources, including soybeans, raw oats. and honey [9,56]. However, the most popular prebiotics are plant oligosaccharides [51]. Nondigestible carbohydrates, including polysaccharides (resistant starch, pectin, and dextrin) and oligosaccharides, such as fructooligosaccharides, galactooligosaccharide, xylooligosacharides, isomaltooligosaccharides, mannanooligosaccharides, raffinose oligosaccharides, arabinoxylanoligosaccharides, lactulose, and inulin, possess prebiotic properties [53,54,57]. Prebiotics have the potential to improve human health by controlling the balance of the intestinal microbiome. They are fermented by the gut bacteria and produce short-chain fatty acids, e.g., propionate, butyrate, and acetate. The production of short-chain fatty acids has positive effects, including improvement of intestinal membrane integrity and absorption of minerals, lowering both glycemic levels and body weight, improved immunity, and modulation of metabolic, cardiovascular, and inflammatory biomarkers [53]. Also, the intake of prebiotics favors the growth of beneficial bacteria, such as Lactobacillus and Bifidobacterium, which are responsible for inhibition of the proliferation of harmful bacteria (Figure 1) [53,54].

Due to the benefits to human health, prebiotics are increasingly used by the food industry as functional ingredients. These compounds can be employed in the production of whole-wheat bread, cereal bars, chocolate, dairy products, infant formulas, and meat products, among others. In addition to natural sources, microorganisms and enzymes can be used for the synthesis of prebiotic compounds [58]. Combinations of prebiotics and probiotics are called synbiotics [53].

## 6. The Influence of Probiotics and Prebiotics on the Mechanisms and Factors Causing CVD

### 6.1. Oxidative Stress

Oxidative stress is known to play a role in the course of CVD [59,60]. This phenomenon refers to elevated intracellular levels of oxygen radicals that cause damage to lipids, proteins, and DNA [61]. Reactive oxygen species (ROS), including superoxide anion radicals, hydroxyl radicals, and hydrogen peroxide, are one of the highly active free radicals. Most living organisms possess enzymatic defenses (superoxide dismutase (SOD), glutathione peroxidase (GPx), glutathione reductase (GR), catalase (CAT), nonenzymatic antioxidant defenses (glutathione (GSH), thioredoxin, vitamin C, vitamin E), and repair systems to protect them against oxidative stress [61]. However, these native antioxidant systems are generally not enough to prevent living organisms from oxidative damage. Many researches showed that probiotic bacteria present significant antioxidant abilities both in vivo and in vitro [62]. ROS can be both endogenously and exogenously generated. Due to their highly reactive nature, ROS can modify other oxygen species, DNA, proteins, or lipids. It is believed that excessive amounts of ROS can cause genomic instability, leading to a variety of chronic diseases, including atherosclerosis and cardiovascular disease [63]. ROS are generated by several enzymatic reactions and chemical processes. NADPH oxidase (NOX) complex is considered to be a major source of ROS generation [64]. There are seven human NOX homologues that function to purposely produce ROS for a range of host defense and signaling functions. Recently, Gómez-Guzmán and colleagues suggested that the probiotics Lactobacillus fermentum CECT5716, Lactobacillus coryniformis CECT5711 (K8), and Lactobacillus gasseri CECT5714 (LC9) (1:1) are able to decrease NOX activity and mRNA expression of NOX-1 and NOX-4 in spontaneously hypertensive rats [65].

Cyclo-oxygenase (COX) is a rate-limiting enzyme in prostaglandin biosynthesis and a two-step enzymatic process in which ROS are generated. COX-2 is upregulated in atherosclerotic lesions and catalyzes the production of the majority of vascular prostanoids in human atherosclerotic areas. Downregulated COX-2 was found in *Helicobacter pylori*-infected mongolian gerbils with a commercial probiotic Lacidofil treatment [66]. Patel and colleagues demonstrated that Lactobacillus acidophilus pretreatment decreased COX-2 expression in catla thymus macrophages compared to Aeromonas hydrophila and co-stimulated macrophages [67].

In recent years, many studies focused on antioxidant properties of probiotics. The culture supernatant, intact cells, and intracellular cell-free extracts of *Bifidobacterium animalis* 01 were found to scavenge hydroxyl radicals and superoxide anions in vitro [61]. Further, oxidative stress in patients with type 2 diabetes was found to be ameliorated by multispecies probiotics [68]. Lactic acid bacteria stains (LAB) were studied widely both in animals and the human body, revealing that LAB can resist ROS, including peroxide radicals, superoxide anions, and hydroxyl radicals [61]. Rats fed high-fat diets supplemented with *Lactobacillus plantarum* P-8 presented an elevated antioxidant ability, as reflected by curtailing the accumulation of liver lipids and protecting healthy liver function [69]. In humans, *Lactobacillus rhamnosus* exerted strong antioxidant activity in situations of elevated physical stress [61]. During the past decades, studies demonstrated that probiotic bacteria strains could exert antioxidant capacity in different ways (Figure 2).

Dietary supplementation of prebiotics, e.g., inulin or oligofructose, contributes to protection from oxidative stress. Inulin, through short-chain fatty acids, can act as a scavenger of reactive oxygen species (ROS). It is also able to modulate responses to pathogenic bacterial insults (LPS) and protect gut from inflammatory processes, probably stimulating defenses against ROS by upregulating colonic mucosal detoxification enzymes (GSTs); in this way, inulin restores the level of some important proteins involved in intestinal smooth muscle contraction [70].

### 6.2. Inflammation

Low-grade inflammation is the cornerstone of many chronic diseases. This type of inflammation increases with age, being common in people of advanced age, and is known to be a risk factor for CVD [6]. In these cardiovascular conditions, higher plasma levels of proinflammatory mediators, such as TNFα, IL1, and IL6, are frequently found. Often, inflammation is linked to an increased intestinal permeability, with elevated intestinal translocation of proinflammatory mediators of bacterial origin, such as LPS. Increased cumulative incidence of CVD with increased serum levels of LPS-binding protein was previously noted. LPS and other bacterial cell membrane constituents are recognized by several receptors on endothelial cells. Binding of LPS directly induces adhesion molecules, such as ICAM-1 and P-selectin on endothelial cells, which are important for interactions with leukocytes [71].

The abovementioned data highlight the potential role of the gut microbiota in controlling intestinal permeability and endotoxemia and, therefore, the development of chronic, low-grade inflammation and the risk for CVD. These findings explain why there is increasing interest in developing intervention strategies targeting the microbiota to achieve downregulation of low-grade inflammation as a way of preventing CVD. Therefore, foods and ingredients, such as probiotics and prebiotics, represent promising tools for the dietary management of CVD risk [6].

Tenorio-Jimenez et al. [72] reported that a 12-week administration of *L. reuteri* V3401 was associated, beyond a reduced risk of CVD, with lower levels of inflammation biomarkers, such as TNF-α, IL-6, IL-8, and soluble intercellular adhesion molecule-1, in obese adults aged 18 to 65 years with metabolic syndrome. However, although some studies demonstrated that probiotics can decrease the production of proinflammatory cytokines, their underlying mechanism remains unclear [9].

In recent years, many research focused on the use of dietary fibers and prebiotics, since many of these polysaccharides can be metabolized by intestinal microbiota, leading to the production of short-chain fatty acids. These metabolites of prebiotic fermentation show anti-inflammatory and immunomodulatory capabilities [53,54]. Kanner et al. [73] showed that inulins, as dietary antioxidants, may play a role in preventing lipid peroxidation in the stomach. In general, dietary supplementation of inulin or oligofructose contributes to protection from oxidative stress, consequently preventing inflammatory reactions associated with oxidative stress [70]. Goderska valuated the prebiotic and anti-inflammatory properties of lactobionic acid (LBA), observing bacterial growth proportional to its concentration, especially for Lactobacilli and Bifidobacterium [74]. LBA is probably resistant to digestive enzymes, so it reaches the colon intact where it is fermented by microbiota. On the other hand, LBA also has anti-inflammatory properties, and it was demonstrated that its administration was associated with a decrease in obesity and better control of metabolic parameters [75].

Isomaltooligosaccarides (IMOs) also promote Lactobacilli and Bifidobacterium growth both in vitro and in vivo [76]. An in vivo study showed the positive effects of isomaltooligosaccarides, green tea extract (GTE), and a combination of IMO and GTE on visceral adipose tissue on the production of proinflammatory cytokines and on lipid and glycemic control. It was also shown to improve insulin, glucagon, and leptin levels in mice [77]. It was investigated that galactooligosaccharides (GOS) can modulate inflammatory process and immune function. GOS increases the levels of cytokine IL-10, interleukin 8 (IL-8), and C-reactive protein and improves natural killer (NK) cell activity [78].

### 6.3. Hypercholesterolemia and High Blood Pressure

Probiotics may reduce cholesterol levels by means of several mechanisms [9]. Most *Bifidobacteria* bacteria demonstrate higher choliloglicin hydrolase activity than do other microorganisms. This enzyme hydrolyzes the amide bonds conjugated with taurine or glycine in bile acids, resulting in the release of primary bile acids; these are easily precipitated at low pH, resulting in their expulsion from the gastrointestinal tract. As these are not reabsorbed from the intestine, they must be replaced by bile produced in the liver from blood cholesterol [51]. Probiotics may exert cholesterol-lowering effects through bile salt hydrolase (an enzyme of probiotics which hydrolyzes bile salts into amino acid residues and free bile salts) [79]. These beneficial effects were demonstrated in both animal models and clinical trials [80,81]. Furthermore, the relationship between gut microbiota, probiotics, and disturbances in lipid metabolism are well explained.

In a randomized, single-blinded, controlled clinical trial, the supplementation with 200 g/day of a probiotic yogurt containing *Streptococcus thermophiles*, *L. bulgaricus*, *L. acidophilus* LA-5, and *B. animalis* BB12 for nine weeks among 70 pregnant women in the third trimester of gestation resulted in a significant reduction in total cholesterol, low-density lipoprotein (LDL) cholesterol, and high-density lipoprotein (HDL) levels, as well as serum triglyceride concentrations [82]. In another study conducted by Hoppu et al. [83], 256 pregnant women allocated into three groups, including dietary counseling with probiotics (*L. rhamnosus* GG and *B. lactis*), placebo dietary counseling, and without counseling (control group) from the first trimester of pregnancy to 12 months postpartum, exhibited similar lipid serum levels during pregnancy. In other research, small-scale, double-blind, placebo-controlled studies observed the beneficial effects of probiotic supplementation in dyslipidemia [84].

It was indicated that probiotic supplementation reduces blood lipid concentrations [85]. Lew et al. [86] found that *L. plantarum* DR7 exerts cholesterol-lowering properties via AMPK phosphorylation. Another group of researchers [87] suggested that the probiotic *L. plantarum* PH40 may also possess cholesterol-lowering properties.

Many clinical trials found probiotic use to be associated with a moderate or significant reduction in blood pressure [9,88,89]. The antihypertensive action of probiotics is believed to act via several mechanisms, including regulating the renin–angiotensin system [89]. Probiotics also play a role in thrombotic disorders. Although the precise roles played by probiotics in the modulation of hemostasis and its various elements, such as blood platelet function, are generally not well documented, several reports were produced on this subject. For example, Schreiber et al. indicated that *L. reuteri* reduces P-selectin expression on the platelet surface and decreases blood platelet–endothelial cell interactions in rats treated with dextran sodium sulfate [90]. Moreover, Haro and Medina [91] reported that the oral administration of *L. casei* CRL431 may be a promising candidate for the prevention of thrombotic complications associated with pneumococcal pneumonia.

In summary, probiotics play an important role in the treatment of various diseases, such as hepatic disease, diarrhea, and gastroenteritis. Probiotics were also shown to have antioxidative, antiplatelet, and anti-inflammatory properties and to lower the level of cholesterol (Figure 3).

Prebiotics also reduce levels of cholesterol. Parnell and Reiner [92] reported that prebiotic intake lowered total serum cholesterol in a hypercholesterolemic rat model. During this research, rats were administered one of three diets with 0, 10, or 20% prebiotic fiber for 10 weeks. Both doses of prebiotic fiber reduced serum cholesterol concentrations about 25%. Moreover, this change was correlated with an increase in caeca digesta, as well as the upregulation of genes involved in cholesterol biosynthesis and bile production. In addition, the obese rats with 10% prebiotic supplementation demonstrated an approximately 40% reduction in triacylglycerol accumulation in the liver. Obesity is often associated with the progression of cardiovascular disease and both probiotic and prebiotic intake were reported to have antiobesogenic effects in various clinical trials [93,94,95].

A number of researches reported synbiotics to possess promising hypercholesterolemic properties [96,97,98]. Mofid et al. [96] noted that the regular intake of synbiotic yogurts reduces the risk of cardiovascular diseases among hypercholesterolemic patients. Liong et al. [97] reported that a synbiotic containing *L. acidophilus* ATCC 4962 reduced total cholesterol, triacylglycerol, and LDL-cholesterol in hypercholesterolemic pigs via the interrelated ways of lipid transporters, including high-density lipoprotein, low-density lipoprotein, and very low-density lipoprotein (VLDL). The animals on the synbiotic diet were fed with *L. acidophilus* ATCC 4962 (1 g/pig per day), mannitol (1.56 g/pig per day), fructo-oligosacharides (1.25 g/pig per day), and inulin (2.2 g/pig per day).

## 7. Conclusions

The intake of probiotics and prebiotics plays an important role in the restoring the normal intestinal flora favoring the growth of beneficial bacteria and reducing the risk of development of chronic ailments, such as cardiovascular disease. Therefore, the interest of these compounds as ingredients for the elaboration of novel foods with functional characteristics is well accepted. Addressing these problems is at the early stages of research. The scientific community must fully clarify how native microbiota affects human health and wellbeing while reliably modeling predictions of interactions of probiotic strains and native gut microbiota, which would allow successful personalization of prebiotic and probiotic therapy, determination of the length of supplementations, and definition of the optimal dosages for individuals to maintain cardiovascular health or to ameliorate some cardiovascular disease.

The field of gut microbiome research is relatively new and complex, and the methods used are far from standardized and harmonized. In many clinical studies, sample sizes were relatively small, with control groups often lacking. A disturbing issue is that different methods were applied for samples for collecting, processing, and storing. It was shown that these methodological differences pose a risk for introducing artefacts. The enormous datasets generated when microbiota, metabolomes, genomes, and transcriptomes are evaluated in cohorts or in intervention studies are challenging, and different complex bioinformatics methods have been applied. There is clearly a need to develop this area of research both in observational and randomized intervention trials. Thus, to enhance the current level of understanding, well-designed clinical trials involving all the aspects of lifestyle, gut microbiota, metabolites, and genetic background should be developed.

## Figures and Tables

**Figure 1 molecules-26-01172-f001:**
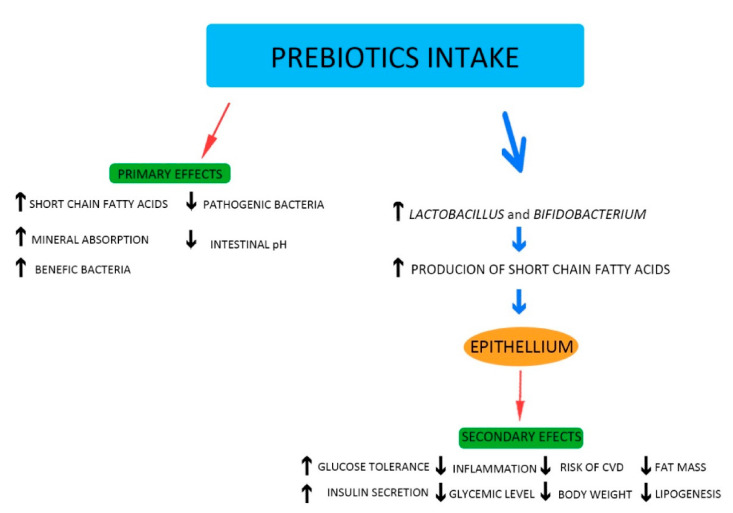
Positive effects of prebiotics.

**Figure 2 molecules-26-01172-f002:**
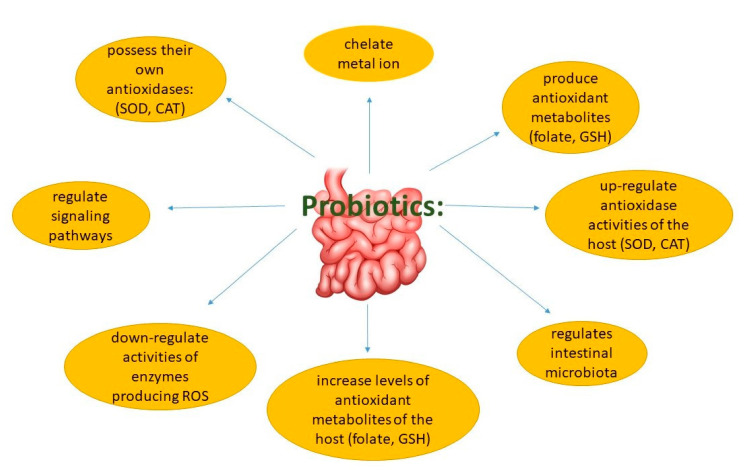
Modulation of antioxidation by probiotics.

**Figure 3 molecules-26-01172-f003:**
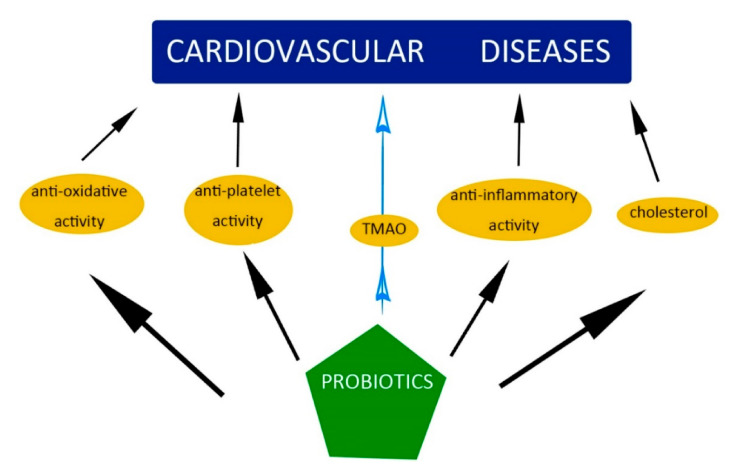
The impact of probiotics on cardiovascular disease (CVD).

**Table 1 molecules-26-01172-t001:** Studies concerning gut dysbiosis in CVD [6].

Study Groups	Microbiota Results	References
Non ischemic heart failure with reduced ejection fraction; *n* = 28 (vs. 19 controls)	↑ *Streptococcus*, *Veillonella*, *Eggerthela* ↓ *Prevotella*, *SMB53* (*Clostridiaceae*)	[42]
Patients with ischemic or dilated cardiomyopathy; *n* = 84 (vs. 266 controls)	↑ *Prevotella*, *Hungatella* (*Lacnospiraceae*), *Succiniclasticum* ↓ *Blautia*, *Anaerostipes*, *Faecalibacterium*, *Lachnospiraceae*, *Bifidobacterium*, *Eubacterium*, *Coprococcus*	[28]
Stable systolic heart failure; *n* = 20 (vs. 20 controls)	↑ *Escherichia-Shigella* ↓ *Blautia*, *Collinsella*, *Ruminococcaceae*, *Erysipelotrichaceaem Faecalibacterium*	[38]
Patients with ischemic or dilated cardiomyopathy; *n* = 53 (vs. 40 controls)	↑ *Ruminococcus*, *Acinetobacter*, *Veillonella* ↓ *Faecalibacterium*, *Alistipes*, *Oscilibacter*	[27]
Patients with hypertension (≥140/90 mmHg); *n* = 60 (vs. 60 controls)	↑ *Klebsiella*, *Salmonella*, *Streptococcus*, *Clostridium*, *Parabacteroides*, *Eggerthella* ↓ *Faecalibacterium*, *Roseburia*, *Synergistetes*	[43]
Patients with hypertension (≥140/90 mmHg) and pre-hypertensive patients (125/80–139/90 mmHg); *n* = 155 (vs. 41 controls)	↑ *Prevotella*, *Klebsiella*, *Porphyromonas* ↓ *Faecalibacterium*, *Roseburia*, *Bifidobacterium*, *Oscillibacter*, *Coprococcus*, *Butyrivibrio*	[44]
Patients with coronary artery disease; *n* = 70 (vs. 98 controls)	↑ *Escherichia-Shigella*, *Lactobacillus*, *Enterococcus*, *Streptococcus* ↓ *Faecalibacterium*, *Roseburia*, *Eubacterium*, *Subdoligranulum*	[45]
Patients with stable angina and old myocardial infarction who underwent percutaneous coronary intervention or bypass; *n* = 39 (vs. 30 controls)	↑ *Lactobacillales* ↓ *Bacteroides*, *Clostridium*	[46]
Patients with atherosclerotic plaques with clinical presentations of stable or unstable angina or acute myocardial infarction; *n* = 218 (vs. 187 controls)	↑ *Enterobacteriaceae*, *Streptococcus*, *Lactobacillus salivarius*, *Atopobium parvulum*, *Ruminococcus gnavus*, *Eggerthella lenta* ↓ *Roseburia*, *Faecalibacterium*	[31]

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
