# Peer review of "Role of Gut Microbiota, Probiotics and Prebiotics in the Cardiovascular Diseases"

_molecules, 2021, doi:10.3390/molecules26041172_

Round 1

Reviewer 1 Report

Dear authors

The review is focused on the effects of prebiotics and probiotics in the prevention and treatment of cardiovascular diseases. However, although the references are recent, there are other reviews in the literature on the same topic. It would therefore be necessary to give an original approach, such as propose future perspectives, molecular mechanisms and possible therapeutic targets rather than a simple bibliographic review.

Author Response

Response for Reviewer 1

The review is focused on the effects of prebiotics and probiotics in the prevention and treatment of cardiovascular diseases. However, although the references are recent, there are other reviews in the literature on the same topic. It would therefore be necessary to give an original approach, such as propose future perspectives, molecular mechanisms and possible therapeutic targets rather than a simple bibliographic review.

The authors would like to thank the Reviewer for the valuable comments which have helped to improve the quality of the paper. All the changes and corrections are highlighted in the text.

We have changed the approach to the topic, according to Reviewer sugestion. We have added section 6. (The influence of probiotics and prebiotics on the mechanisms and factors causing CVD) in which we described the most important mechanisms contributing to CVD development (oxidative stress, inflammation, hypercholesterolemia and high blood pressure) and influence of probiotics and prebiotics on these mechanisms and factors. In the section 3. we presented diet-derived metabolic products produced by gut microbiota, capable of influencing the host’s cardiovascular conditio. Moreover, we supplemented the conclusions with caveats in research and future prospects.

Reviewer 2 Report

  • The review needs to be improved for publication. In my opinion, each subtitle must be reorganized and detailed. For example, when talking about inflammation or oxidative stress you need to emphasize this processes in cardiovascular disease. After that, you need to insist on the pre and probiotic role in them.
  • Does the microbiota have the same impact on all cardiovascular diseases? It is recommended to talk about the most common cardiovascular diseases, their pathophysiology and then about the pre- and probiotic impact.

Author Response

Response for Reviewer 2

The review needs to be improved for publication. In my opinion, each subtitle must be reorganized and detailed. For example, when talking about inflammation or oxidative stress you need to emphasize this processes in cardiovascular disease. After that, you need to insist on the pre and probiotic role in them.

Does the microbiota have the same impact on all cardiovascular diseases? It is recommended to talk about the most common cardiovascular diseases, their pathophysiology and then about the pre- and probiotic impact.

The authors would like to thank the Reviewer for the valuable comments which have helped to improve the quality of the manuscript. All the changes and corrections are highlighted in the text.

We have reorganized the review, according to Reviewer sugestion. We have added section 6. (The influence of probiotics and prebiotics on the mechanisms and factors causing CVD) in which we described the most important mechanisms contributing to CVD development (oxidative stress, inflammation, hypercholesterolemia and high blood pressure) and influence of probiotics and prebiotics on these factors and diseases. In the section 3. we presented diet-derived metabolic products produced by gut microbiota, capable of influencing the host’s cardiovascular condition. Moreover, we supplemented our review with section 2. entitled “Cardiovascular diseases - risk factors and the mechanism of development”.

Reviewer 3 Report

This is a very nice review of the influence of the gut microbiota on cardiovascular disease.  The only comment/suggestion for improvement would be to include mechanisms of action for probiotics and prebiotics on the various parameters of CVD.  For example, do the authors have information regarding the role of nitric oxide, prostaglandins, etc... as a mechanism of action?

Author Response

Response for Reviewer 3

This is a very nice review of the influence of the gut microbiota on cardiovascular disease.  The only comment/suggestion for improvement would be to include mechanisms of action for probiotics and prebiotics on the various parameters of CVD.  For example, do the authors have information regarding the role of nitric oxide, prostaglandins, etc... as a mechanism of action?

The authors would like to thank the Reviewer for appreciating the work and customs comments. All the changes and corrections are highlighted in the text.

We have reorganized the review, according to Reviewer sugestion. We have added section 6. (The influence of probiotics and prebiotics on the mechanisms and factors causing CVD) in which we described the most important mechanisms contributing to CVD development (oxidative stress, inflammation, hypercholesterolemia and high blood pressure) and influence of probiotics and prebiotics on these factors and diseases. In the section 3. we presented diet-derived metabolic products produced by gut microbiota, capable of influencing the host’s cardiovascular condition. Moreover, we supplemented our review with section 2. entitled “Cardiovascular diseases - risk factors and the mechanism of development”.

Round 2

Reviewer 2 Report

Your manuscript has been improved.